# Enhancement in Power Conversion Efficiency of Perovskite Solar Cells by Reduced Non-Radiative Recombination Using a Brij C10-Mixed PEDOT:PSS Hole Transport Layer

**DOI:** 10.3390/polym15030772

**Published:** 2023-02-02

**Authors:** Sehyun Jung, Seungsun Choi, Woojin Shin, Hyesung Oh, Jaewon Oh, Mee-Yi Ryu, Wonsik Kim, Soohyung Park, Hyunbok Lee

**Affiliations:** 1Department of Physics and Institute of Quantum Convergence Technology, Kangwon National University, 1 Gangwondaehak-gil, Chuncheon-si 24341, Republic of Korea; 2Advanced Analysis Center, Korea Institute of Science and Technology, 5 Hwarang-ro 14-gil, Seongbuk-gu, Seoul 02792, Republic of Korea

**Keywords:** perovskite solar cell, PEDOT:PSS, Brij C10, hole transport layer, non-radiative recombination

## Abstract

Interface properties between charge transport and perovskite light-absorbing layers have a significant impact on the power conversion efficiency (PCE) of perovskite solar cells (PSCs). Poly(3,4-ethylenedioxythiophene):poly(styrene sulfonate) (PEDOT:PSS) is a polyelectrolyte composite that is widely used as a hole transport layer (HTL) to facilitate hole transport from a perovskite layer to an anode. However, PEDOT:PSS must be modified using a functional additive because PSCs with a pristine PEDOT:PSS HTL do not exhibit a high PCE. Herein, we demonstrate an increase in the PCE of PSCs with a polyethylene glycol hexadecyl ether (Brij C10)-mixed PEDOT:PSS HTL. Photoelectron spectroscopy results show that the Brij C10 content becomes significantly high in the HTL surface composition with an increase in the Brij C10 concentration (0–5 wt%). The enhanced PSC performance, e.g., a PCE increase from 8.05 to 11.40%, is attributed to the reduction in non-radiative recombination at the interface between PEDOT:PSS and perovskite by the insulating Brij C10. These results indicate that the suppression of interface recombination is essential for attaining a high PCE for PSCs.

## 1. Introduction

Perovskite solar cells (PSCs) are considered a promising renewable energy source owing to their low fabrication cost, tunable band gap, high light absorbance, and mechanical flexibility [1,2,3,4,5,6,7]. To increase the power conversion efficiency (PCE), charge transport layers are inserted between the perovskite light-absorbing layer and the electrodes [8,9,10,11]. SnO_2_, TiO_2_, and [6,6]-phenyl C_61_ butyric acid methyl ester (PCBM) have been employed as electron transport layers [12,13]. Meanwhile, 2,2′,7,7′-tetrakis[*N*,*N*-di(4-methoxyphenyl)amino]-9,9′-spirobifluorene (spiro-OMeTAD), copper(I) thiocyanate (CuSCN), and poly(3,4-ethylenedioxythiophene)-poly(styrene sulfonate) (PEDOT:PSS) have been used as a hole transport layers (HTLs) [14,15]. Among them, PEDOT:PSS is a polyelectrolyte composite consisting of positively charged PEDOT and negatively charged PSS. In a PEDOT:PSS film, conducting PEDOT chains are covered with insulating PSS chains [16,17,18]. The PEDOT:PSS HTL is popularly utilized owing to its high work function, high hole transport ability, and water processability.

Recently, instead of neat PEDOT:PSS, modified PEDOT:PSS with other functional materials has been widely investigated because of its efficient anode contact formation in PSCs [19,20]. Doping of PEDOT:PSS is an effective method to adjust electronic properties. For example, 2,3,5,6-tetrafluoro-7,7,8,8-tetracyanoquinodimethane (F4-TCNQ)-doped PEDOT:PSS has a greater work function and increased conductivity, thereby increasing the PCE of PSCs [21]. The addition of sulfonic acid-functionalized graphene oxides increases the work function of PEDOT:PSS, forming a favorable energy level alignment [22]. The CuSCN-doped PEDOT:PSS simultaneously improves the hole extraction efficiency and device lifetime [23]. Such doping improves the hole transport ability of PEDOT:PSS, thereby improving PSC performance.

In contrast, the high electrical conductivity of PEDOT:PSS causes unwanted non-radiative recombination. Therefore, the reduction in recombination is another valuable strategy for increasing the PCE [24]. Inserting an ultrathin insulating layer between the perovskite and PEDOT:PSS layers reduces interface recombination. For example, the poly(methyl methacrylate) (PMMA) interlayer significantly increases the PCE of PSCs by suppressing recombination [25]. The deposition of phenylethylammonium bromide (PEABr) onto PEDOT:PSS generates a low-dimensional perovskite interlayer, thereby decreasing trap-state density and increasing the size of perovskite crystals [26]. The mixing of a non-ionic surfactant polyethylene glycol *tert*-octylphenyl ether (Triton X) with PEDOT:PSS forms a Triton X-rich surface, and the PCE of PSCs is remarkably increased by reduced recombination [27]. Non-ionic surfactants also improve the wetting of PEDOT:PSS; therefore, their use could be more effective than other insulating materials for achieving efficient PSCs.

Polyethylene glycol hexadecyl ether (Brij C10) is also a famous non-ionic surfactant. We previously reported that the Brij C10 additive significantly decreases the sheet resistance of a highly conducting PEDOT:PSS (Clevios PH1000) anode, thereby improving the PCE of PSCs [28]. However, the working mechanism of the Brij C10 additive in semiconducting PEDOT:PSS (Clevios P VP AI 4083) and its role in the HTL would be different. In this regard, here, we investigated the effects of the Brij C10 additive on the PEDOT:PSS HTL for enhancing the PCE of PSCs. PSCs with various concentrations of the Brij C10 additive were characterized. The electronic structures and morphological properties of the Brij C10-mixed PEDOT:PSS films were analyzed. Finally, the origin of the increased PCE was explained by the reduction in non-radiative recombination at the interface between PEDOT:PSS and perovskite. This study provides a fundamental understanding of designing a PEDOT:PSS HTL modified with an additive.

## 2. Experimental Methods

The PSCs were fabricated on indium tin oxide (ITO)-coated glass substrates (AMG). ITO substrates were cleaned through ultrasonication in deionized (DI) water, detergent, acetone, methanol, and DI water, and subsequently dried under a N_2_ gas flow. Afterwards, the substrates were treated with ultraviolet–ozone (UV–O_3_) for 15 min at 100 °C, utilizing a PSDP-UV4T UV–O_3_ cleaner (Novascan Technology Inc., Boone, IA, USA). A PEDOT:PSS (Clevios P VP AI 4083, Heraeus GmbH, Hanau, Germany) solution was mixed with various concentrations (0, 1, 2, 3, 4, and 5 wt%) of Brij C10 (average M_n_: ~683, Sigma Aldrich Inc., St. Louis, MO, USA). The Brij C10-mixed PEDOT:PSS solution was stirred overnight at room temperature (RT) before use. The Brij C10-mixed PEDOT:PSS film was deposited with spin coating at two step spin rates of 500 rpm for the initial 5 s and 3000 rpm for the latter 30 s on the cleaned ITO substrate. Subsequently, the samples were annealed at 150 °C for 10 min using a hot plate. The p-i-n structure PSCs consisted of Ag (70 nm)/PCBM (60 nm)/perovskite (~550 nm)/PEDOT:PSS/ITO. The triple cation perovskite films were fabricated with a one-step spin-coating method. PbBr_2_ (purity = 99.999%, Sigma-Aldrich Inc., St. Louis, MO, USA), PbI_2_ (purity = 99.99%, Tokyo Chemical Industry Co., Ltd., Tokyo, Japan), CsI (purity > 99.0%, Tokyo Chemical Industry Co., Ltd., Tokyo, Japan), methylammonium bromide (MABr, purity > 99.99%, Greatcell Solar Materials Pty Ltd., Queanbeyan, Australia), and formamidinium iodide (FAI, purity > 99.99%, Greatcell Solar Materials Pty Ltd., Queanbeyan, Australia) weighing 159.6, 1136.4, 132, 24.4, and 374 mg, respectively, were dissolved in 2 mL of a mixed solvent of *N*,*N*-dimethylformamaide (DMF, purity = 99.8%, Sigma-Aldrich Inc., St. Louis, MO, USA):dimethyl sulfoxide (DMSO, purity ≥ 99.5%, Sigma-Aldrich Inc., St. Louis, MO, USA) with a volume ratio of 4:1, and the solution was stirred overnight at RT. The perovskite precursor solution (Cs_0.18_FA_0.75_MA_0.08_PbI_2.63_Br_0.38_) was spin-coated onto the Brij C10-mixed PEDOT:PSS/ITO at two step spin rates of 500 rpm for 5 s followed by 5000 rpm for 45 s. Chlorobenzene (CB, purity > 98.0%, Tokyo Chemical Industry Co., Ltd., Tokyo, Japan) anti-solvent of 0.3 mL was dropped onto the center of the sample at 15 s before the end of the spinning. All the perovskite layers were carefully fabricated with the same methods and conditions, except for the Brij C10 concentration in the underlying PEDOT:PSS layer. Subsequently, the perovskite films were annealed at 100 °C for 45 min. PCBM (purity > 99.5%, Luminescence Technology Co.) was dissolved in CB (purity = 99.9%, Sigma-Aldrich Inc., St. Louis, MO, USA) at a concentration of 20 mg mL^−1^ and stirred overnight at 80 °C. The PCBM solution was filtered through a 0.45 μm polytetrafluorethylene membrane and spin-coated on the perovskite/Brij C10-mixed PEDOT:PSS/ITO at a spin rate of 1500 rpm for 40 s. The Brij C10-mixed PEDOT:PSS layer was deposited under ambient conditions, whereas the perovskite and PCBM layers were deposited in a N_2_-filled glove box. Finally, Ag (purity = 99.99%, Taewon Science Co., Ltd., Seoul, Republic of Korea) was deposited onto the sample in a high vacuum chamber via thermal evaporation with deposition rates of 0.01 nm s^−1^ for the initial 15 nm and 0.05 nm s^−1^ for the subsequent 55 nm. The device area was 2 mm × 2 mm.

The PSC performance was recorded using a Keithley 2400 source measure unit (Tektronix Inc., Beaverton, OR, USA) under air mass (AM) 1.5G 1 sun illumination generated by a Model 10,500 solar simulator (Abet Technologies Inc., Milford, CT, USA). Ultraviolet and X-ray photoelectron spectroscopy (UPS and XPS) measurements were conducted using a Nexsa G2 spectrometer (Thermo Fisher Scientific Inc., Waltham, MA, USA). UPS and XPS spectra were collected using a He I_α_ discharge lamp (hν = 21.22 eV) and an Al K_α_ X-ray source (hν = 1486.6 eV), respectively. To obtain the secondary electron cutoff (SEC), we applied a bias of −10 V. Atomic force microscopy (AFM) images were obtained using Nanoscope V MultiMode 8 (Bruker Co., Billerica, MA, USA) operated in tapping mode for a 2 μm × 2 μm analysis area. Scanning electron microscopy (SEM) images were obtained using an S-4800 microscope (Hitachi High-Tech Co., Tokyo, Japan). Steady-state photoluminescence (PL) spectra were obtained using a Si photodetector (Electro-Optical Systems Inc., Phoenixville, PA, USA) with a 532 nm diode pump solid-state laser (Changchun New Industries Optoelectronics Technology Co., Changchun, China) as the excitation light. Time-resolved PL spectra were obtained by using an FLS 920 fluorescence spectrometer (Edinburgh Instruments Ltd., Livingston, United Kingdom). A 656 nm picosecond-pulsed diode laser (EPL-655, Edinburgh Instruments Ltd., Livingston, United Kingdom) was used for excitation in the time-resolved PL measurements. Although one sample for each Brij C10 concentration was fabricated for analyzing the electronic and structural characteristics, it was sufficient to observe remarkable changes in the material properties. All the samples were fabricated upon the ITO-coated glass substrates with the size of 1.5 cm × 1.5 cm. Experientially, perovskite and organic films deposited on those substrates show high uniformity. Therefore, the errors associated with different analysis positions would not be significant.

## 3. Results and Discussion

Figure 1a shows the device structure of PSCs and the chemical structure of Brij C10. Figure 1b shows the J–V characteristics of the best PSCs with a PEDOT:PSS HTL of various concentrations of the Brij C10 additive under AM 1.5G 1 sun illumination. The measured solar cell parameters and statistics evaluated from 10 devices are presented in Table 1. The PSC with the pristine PEDOT:PSS HTL shows a short-circuit current density (J_SC_) of 17.75 mA cm^−2^, an open-circuit voltage (V_OC_) of 0.85 V, and a fill factor (FF) of 53.2%, resulting in a PCE of 8.05%. However, all solar cell parameters were enhanced with the addition of Brij C10 into PEDOT:PSS. The PSC with 1 wt% Brij C10-mixed PEDOT:PSS HTL shows a J_SC_ = 18.62 mA cm^−2^, a V_OC_ = 0.90 V, an FF = 59.7%, and a PCE = 10.00%. The PCE of PSCs increased further with the 2 wt% Brij C10-mixed PEDOT:PSS HTL, exhibiting a J_SC_ = 19.08 mA cm^−2^, a V_OC_ = 0.91 V, an FF = 60.2%, and a PCE = 10.45%. The highest PCE was achieved with the 3 wt% Brij C10-mixed PEDOT:PSS HTL, with a J_SC_ = 20.61 mA cm^−2^, a V_OC_ = 0.89 V, an FF = 62.0%, and a PCE = 11.40%. However, further increases in the Brij C10 concentration deteriorated the PCE. The PSC of the 4 wt% Brij C10-mixed PEDOT:PSS HTL shows a J_SC_ = 19.51 mA cm^−2^, a V_OC_ = 0.90 V, an FF = 57.3%, and a PCE = 10.04%. Moreover, the PSC of the 5 wt% Brij C10-mixed PEDOT:PSS HTL shows a J_SC_ = 19.26 mA cm^−2^, a V_OC_ = 0.91 V, an FF = 50.0%, and a PCE = 8.80%. Figure 1c–f shows the box plots of J_SC_, V_OC_, FF, and PCE. The statistics show the same trend as the best PCE of PSCs, indicating that the device results were reliably obtained.

To understand the changes in the valence electronic structure of PEDOT:PSS films with the Brij C10 additive, UPS measurements were performed. Figure 2 shows the UPS spectra of the SEC and highest occupied molecular orbital (HOMO) regions of PEDOT:PSS films on ITO with various concentrations of the Brij C10 additive. The SEC region spectra were normalized for clear comparison. The SEC drawn with the kinetic energy scale indicates the work function of the sample. In the pristine PEDOT:PSS (0 wt%), the work function of PEDOT:PSS was as high as 5.38 eV. The work function of P VP AI 4083 for HTL was higher than that of PH1000 for an electrode, owing to the higher PSS/PEDOT ratio [28,29,30]. However, the work function gradually decreased with Brij C10 mixing. For the Brij C10 concentrations of 1, 2, 3, 4, and 5 wt%, the work function decreased to 4.96, 4.64, 4.45, 4.47, and 4.32 eV, respectively. This work function reduction might stem from an interface dipole due to the Brij C10 on the surface [31]. The slight fluctuation in the work function at concentrations of 3 and 4 wt% might be attributed to the inhomogeneity in surface composition. In general, the high work function of a HTL decreases the energy offset between the HOMO level of a HTL and valence band maximum (VBM) of a perovskite layer [32]. However, the 1–3 wt% Brij C10 additives decreased the work function of PEDOT:PSS by 0.42–0.93 eV; hence, the energy offset could not be reduced. Therefore, the changes in the work function cannot be the origin of the enhanced PCEs with the 1–3 wt% Brij C10 additives.

In the HOMO region, characteristic peaks of the pristine PEDOT:PSS were observed at 3.0, 5.3, and 8.0 eV. However, these peaks rapidly attenuate with the addition of the Brij C10 additive. With the minimum mixture of 1 wt%, most spectral features originate from Brij C10. The Brij C10 features located at 4.0 and 6.4 eV develop with an increase in the Brij C10 concentration. This indicates that the sample surface mostly consists of Brij C10, agreeing well with the XPS results in Figure 3, Figure 4 and Figure 5.

XPS measurements were performed to investigate the core-level electronic structure. Figure 3a shows the S 2p XPS spectra of Brij C10-mixed PEDOT:PSS films on ITO. The S 2p peaks of PEDOT and PSS are clearly distinguished by different energy positions [33,34,35,36,37]. The lower binding energy peaks originate from the PEDOT moiety, whereas the higher binding energy peaks originate from the PSS moiety. All the S 2p peak intensities decreased as Brij C10 was mixed. This indicates that the surface composition of PEDOT:PSS decreases. Figure 3b shows the fittings of the measured S 2p XPS spectra. In the XPS fittings, we followed the interpretation of a previous study performed by Greczynski et al. [33]. Each component has two S 2p_3/2_ and 2p_1/2_ peaks with an energy difference of 1.2 eV due to spin–orbit splitting. In the pristine PEDOT:PSS, S_I_ peaks at 163.7 and 164.9 eV from the S–C bond in PEDOT, S_II_ peaks at 167.6 and 168.8 eV from the S(–ONa)=O bond in PSS, and S_III_ peaks at 168.1 and 169.3 eV from the S(–OH)=O bond in PSS were observed. The S_I_ peak is asymmetric owing to delocalized positive charges in conjugated thiophene rings [33]. The peak positions shift toward higher binding energies with an increase in the Brij C10 concentration, similar to the SEC shifts. With 5 wt% Brij C10 additive, the S_I_, S_II_, and S_III_ peaks are located at 164.0 and 165.2 eV, 168.1 and 169.3 eV, and 168.9 and 170.1 eV, respectively. The energy difference between the SEC and core-level shifts is due to the difference in probing depth between UPS and XPS.

Figure 4a shows the O 1s XPS spectra of the Brij C10-mixed PEDOT:PSS films on ITO. In the pristine PEDOT:PSS, several oxygen peaks are convoluted. However, only one peak at high binding energies becomes dominant as Brij C10 is added. Figure 4b shows the fittings of the measured O 1s XPS spectra. Four components are assigned to the pristine PEDOT:PSS; that is, the O_I_ peak at 531.1 eV from the Na–O–S bond in PSS, the O_II_ peak at 531.6 eV from the O=S bond in PSS, the O_III_ peak at 532.6 eV from the H–O–S bond in PSS, and the O_IV_ peak at 532.9 eV from the O–C bond in PEDOT and PSS. The O_III_ peak is known to be broader than the other peaks owing to the hydrogen bond [33]. Similar to the S 2p peaks, the O 1s peaks shift toward higher binding energies as Brij C10 is added. The intensities of the O_I_, O_II_, and O_III_ peaks decrease, whereas that of the O_IV_ peak increases with an increase in the Brij C10 concentration owing to the O–C bond in Brij C10. With 5 wt% Brij C10, only the O_IV_ peak barely remains. This is interpreted as a result of the high content of Brij C10 on the film surface.

Figure 5a shows the C 1s XPS spectra of Brij C10-mixed PEDOT:PSS films on ITO and Figure 5b shows their fittings. In the pristine PEDOT:PSS, three components are observed: the C_I_ peak at 283.9 eV from the C–C bond, the C_II_ peak at 284.4 eV from the C–S bond, and the C_III_ peak at 285.8 eV from the C–O bond. As Brij C10 is added, the C_II_ peak intensity decreases, whereas the C_III_ peak intensity increases. Brij C10 also contains a C–C bond; therefore, the C_I_ peak intensity does not significantly change. With 5 wt% Brij C10, both the C_I_ and C_III_ peaks become dominant. The C 1s peaks shift toward higher binding energies such as the valence level and other core-level shifts. Connecting all the XPS results, the surface of the Brij C10-mixed PEDOT:PSS film is primarily composed of Brij C10. This is similar to the case of Triton X in the PEDOT:PSS HTL [27]. Therefore, Brij C10 would play the same role as Triton X at the interface between perovskite and PEDOT:PSS in PSCs.

Figure 6 shows the AFM images of Brij C10-mixed PEDOT:PSS films on ITO. In the pristine PEDOT:PSS film, a typical small granular structure with high uniformity is observed [38,39,40]. The root mean square roughness (R_RMS_) of the pristine PEDOT:PSS film is 0.9 nm. However, the sample surface is roughened and many voids appear as Brij C10 is added. Consequently, the R_RMS_ increases with an increase in the Brij C10 concentration. The measured R_RMS_ values of 1, 2, 3, 4, and 5 wt% Brij C10-mixed PEDOT:PSS films are 2.4, 3.5, 4.2, 4.8, and 5.2 nm, respectively. Such changes in surface features are the same as observed in PH1000 with Brij C10 [28]. In addition, the film thickness might be similarly increased with the changes in the Brij C10 concentration. These results show that the surface uniformity of the PEDOT:PSS film degrades with the Brij C10 additive. The roughened surface would deteriorate the morphology of an upper-lying perovskite layer, which is detrimental to PSC performance. Therefore, the morphological properties do not contribute to the increased PCE of PSCs with 1–3 wt% Brij C10 concentrations; other mechanisms play a decisive role in device performance.

To investigate the changes in the morphology of perovskite films with the underlying Brij C10-mixed PEDOT:PSS HTL, SEM images were obtained for the perovskite/Brij C10-mixed PEDOT:PSS/ITO samples. Figure 7 shows the top-view SEM images of perovskite films deposited on the Brij C10-mixed PEDOT:PSS/ITO and the statistics of their grain size. The grain size was evaluated by counting the length of each grain in eight different directions. The device area is much larger than the analysis area; therefore, the average size influences its device performance. In all SEM images, perovskite crystals are clearly observed, indicating that perovskite was formed appropriately. However, the crystal size gradually decreases as Brij C10 is added. On the pristine PEDOT:PSS HTL, the grain size of perovskite was evaluated as 191 ± 78 nm. With 1, 2, 3, 4, and 5 wt% Brij C10 concentrations, the grain size decreases to 159 ± 49, 135 ± 45, 125 ± 48, 121 ± 42, and 99 ± 40 nm, respectively. This is attributed to the roughened surface of the underlying PEDOT:PSS layer with the Brij C10 additive, as expected in the AFM images (Figure 6). In general, larger perovskite grains result in a higher PCE owing to superior charge transport, lower trap density, and reduced bulk recombination [41,42,43]. Therefore, the changes in the grain size cannot explain the improvement in the PCE of PSCs with Brij C10, and other mechanisms govern carrier and exciton behaviors in such devices.

The changes in the exciton dynamics of perovskite on the Brij C10-mixed PEDOT:PSS HTL were explored with PL measurements. Figure 8a shows the steady-state PL spectra of perovskite films on glass, ITO, pristine PEDOT:PSS (0 wt%)/ITO, and 3 wt% Brij C10-mixed PEDOT:PSS/ITO. The concentration of 3 wt% was selected because it showed the highest PCE among the fabricated PSCs. For clarity, the PL intensities of perovskite films on the glass and ITO substrates were drawn with 0.1 times the original intensities. The PL peak was observed at 764 nm, indicating a band gap of 1.62 eV. The PL intensity was lower on ITO than on glass due to hole transport to ITO. The PL intensity significantly decreases on the PEDOT:PSS/ITO, indicating superior hole transport. However, the PL intensity distinctly increases in 3 wt% Brij C10-mixed PEDOT:PSS. This is caused by a reduced non-radiative recombination at the interface of perovskite and HTL. Nonetheless, the PL intensity of perovskite/3 wt% Brij C10-mixed PEDOT:PSS is still significantly lower than that of perovskite/ITO, implying that efficient hole transport is maintained.

Figure 8b shows the time-resolved PL spectra of perovskite films on glass, ITO, pristine PEDOT:PSS/ITO, and 3 wt% Brij C10-mixed PEDOT:PSS/ITO. The perovskite films on both glass and ITO show very slow decay. In contrast, the PL intensity decay becomes very fast with the insertion of the PEDOT:PSS HTL between perovskite and ITO. Note that the PL intensity decay becomes slower with the addition of 3 wt% Brij into the PEDOT:PSS HTL. To evaluate the decay time, the PL intensity curve was fitted with tri-exponential functions:$$I\left(t\right)=A_{1}e^{-\frac{t}{\tau_{1}}}+A_{2}e^{-\frac{t}{\tau_{2}}}+A_{3}e^{-\frac{t}{\tau_{3}}}$$
where I(t) is the PL intensity, A_n_ is a pre-exponential factor, t is time, and τ_n_ is a time constant (n = 1, 2, and 3). The average τ (τ_avg_) is calculated as follows:$$\tau_{\mathrm{avg}}=\frac{A_{1}\tau_{1}^{2}+A_{2}\tau_{2}^{2}+A_{3}\tau_{3}^{2}}{A_{1}\tau_{1}+A_{2}\tau_{2}+A_{3}\tau_{3}}$$

The fitting parameters are listed in Table 2. A_1_, τ_1_, A_2_, and τ_2_ are the fast decay components originating from charge transport and non-radiative recombination, whereas A_3_ and τ_3_ are the slow decay components originating from radiative recombination. On the glass substrate, only slow decay is dominant (99.27%); hence, τ_avg_ becomes as high as 464.77 ns. On the ITO substrate, the portion of the slow decay decreases (90.57%) owing to hole transport to ITO; thus, τ_avg_ decreases to 370.16 ns. However, efficient hole transport occurs with the insertion of the PEDOT:PSS HTL. Thereby, the portion of the fast decay increases (37.70%), whereas that of the slow decay decreases (62.30%), resulting in a low τ_avg_ of 27.43 ns. In contrast, with 3 wt% Brij C10-mixed PEDOT:PSS HTL, the portion of the fast decay decreases (36.31%), whereas that of the slow decay increases (63.69%); consequently, τ_avg_ increases to 30.30 ns. Such time-resolved PL analysis is in excellent agreement with the steady-state PL analysis. These PL results indicate that unwanted non-radiative recombination is suppressed by the insulating Brij C10, whereas efficient hole transport is retained. This is the origin of the enhanced PCE of PSCs with the Brij C10-mixed PEDOT:PSS HTL. The reduction in interface recombination plays a crucial role in increasing the PCE of PSCs [27,44,45,46]. However, the insertion of Brij C10 with concentrations higher than 3 wt% into PEDOT:PSS renders the HTL more insulating; thus, holes cannot be transported efficiently. The worsened morphology of perovskite by Brij C10 also deteriorates PSC performance. Therefore, an optimum amount of Brij C10 should be mixed with PEDOT:PSS for increasing the PCE.

## 4. Conclusions

In this study, we investigated the effects of the Brij C10 additive on the semiconducting PEDOT:PSS HTL in PSCs. The PCE of PSCs markedly increases from 8.05% with the pristine PEDOT:PSS HTL to 11.40% with the 3 wt% Brij C10-mixed PEDOT:PSS HTL. UPS and XPS results show that the surface of the PEDOT:PSS HTL contains a high content of Brij C10. Steady-state and time-resolved PL results show that non-radiative recombination is significantly reduced by the insulating Brij C10 on the PEDOT:PSS HTL surface. The suppressed interface recombination enhances the PCE of PSCs despite the reduced work function of PEDOT:PSS and the reduced size of the perovskite crystals. Therefore, the addition of a non-ionic surfactant is an effective method to weaken exciton quenching at the PEDOT:PSS interface in PSCs for increasing the PCE.

## Figures and Tables

**Figure 1 polymers-15-00772-f001:**
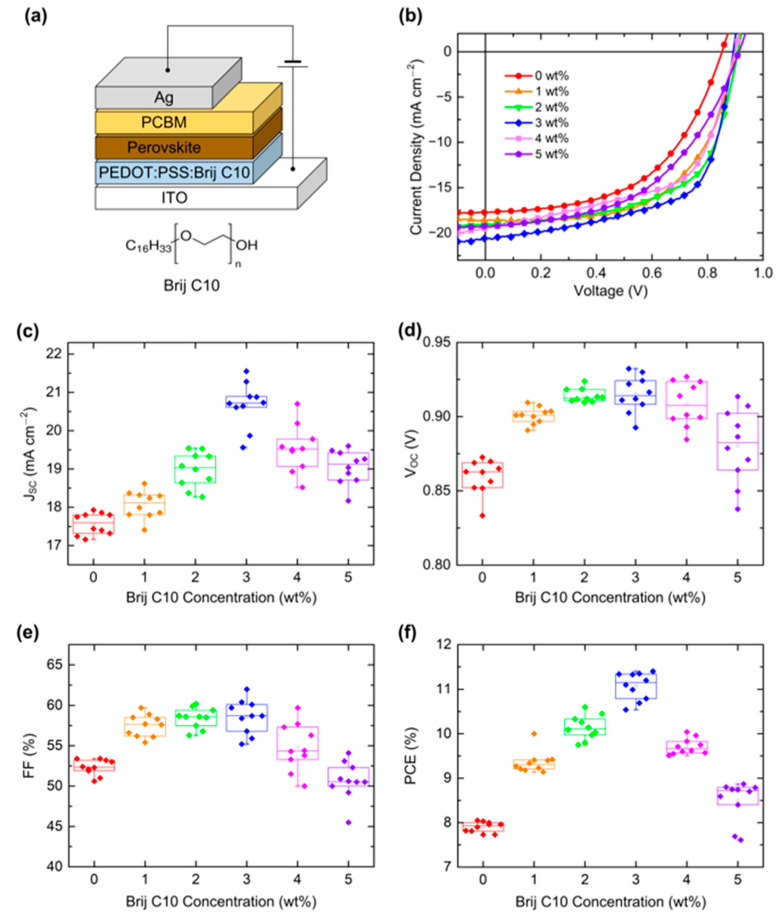
(**a**) Device structure of PSCs and chemical structure of Brij C10. (**b**) J–V characteristics of PSCs with Brij C10-mixed PEDOT:PSS HTL under AM 1.5G 1 sun illumination. Box plots of (**c**) J_SC_, (**d**) V_OC_, (**e**) FF, and (**f**) PCE of PSCs. The data were obtained from 10 devices.

**Figure 2 polymers-15-00772-f002:**
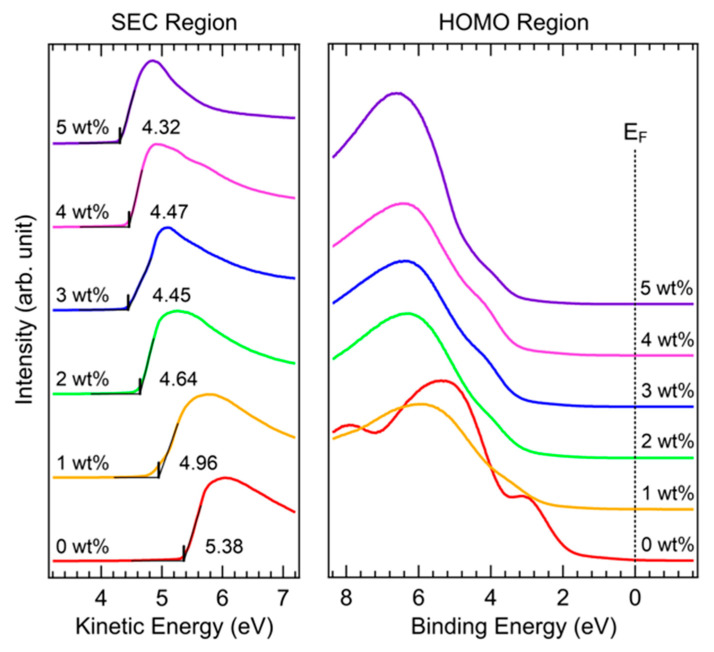
UPS spectra of SEC and HOMO regions of PEDOT:PSS films on ITO with various concentrations of Brij C10 additive (0, 1, 2, 3, 4, and 5 wt%).

**Figure 3 polymers-15-00772-f003:**
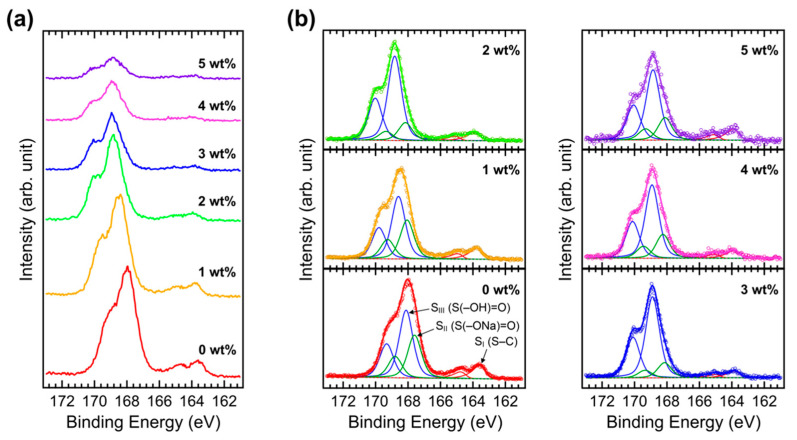
(**a**) XPS S 2p spectra of PEDOT:PSS films on ITO with various concentrations of Brij C10 additive (0, 1, 2, 3, 4, and 5 wt%) and (**b**) their fittings.

**Figure 4 polymers-15-00772-f004:**
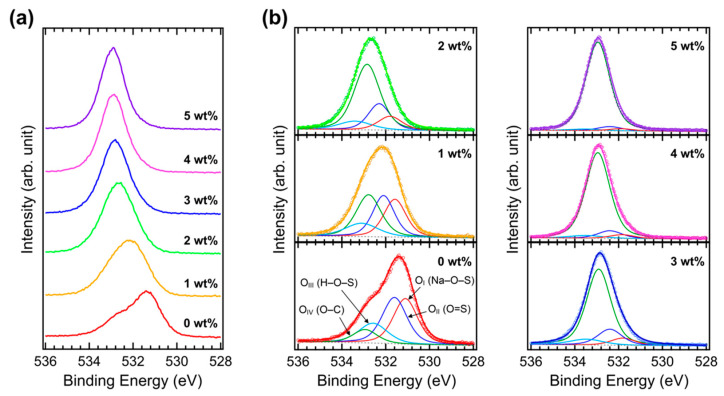
(**a**) XPS O 1s spectra of PEDOT:PSS films with various concentrations of Brij C10 additive (0, 1, 2, 3, 4, and 5 wt%) and (**b**) their fittings.

**Figure 5 polymers-15-00772-f005:**
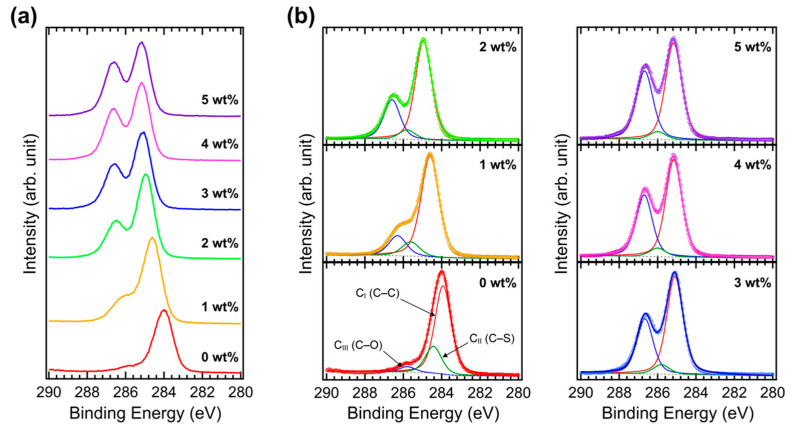
(**a**) XPS C 1s spectra of PEDOT:PSS films with various concentrations of Brij C10 additive (0, 1, 2, 3, 4, and 5 wt%) and (**b**) their fittings.

**Figure 6 polymers-15-00772-f006:**
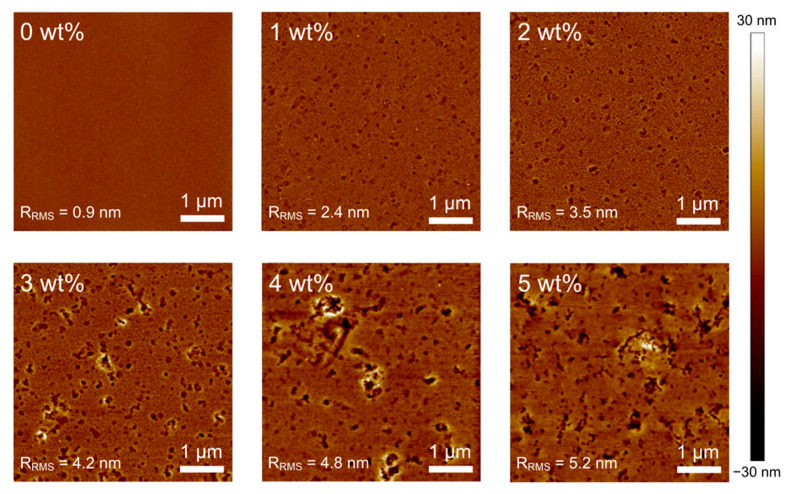
AFM images of PEDOT:PSS films with various concentrations of Brij C10 additive (0, 1, 2, 3, 4, and 5 wt%).

**Figure 7 polymers-15-00772-f007:**
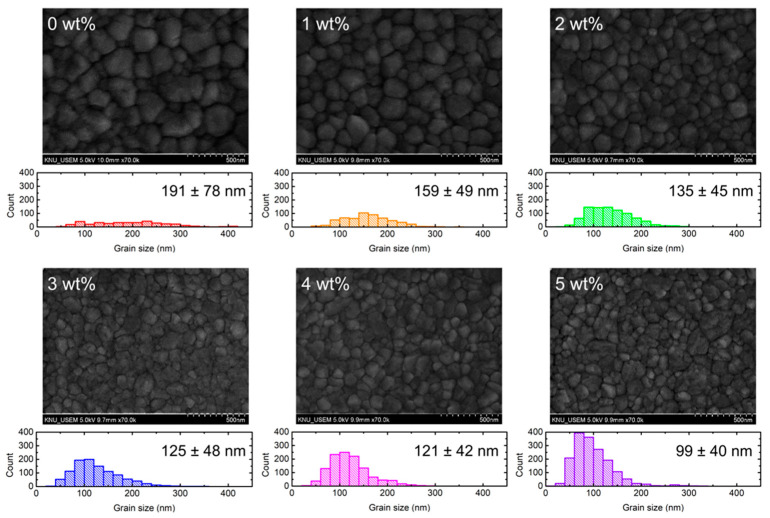
Top-view SEM images of perovskite film on Brij C10-modified (0, 1, 2, 3, 4, and 5 wt%) PEDOT:PSS/ITO and statistics of grain size.

**Figure 8 polymers-15-00772-f008:**
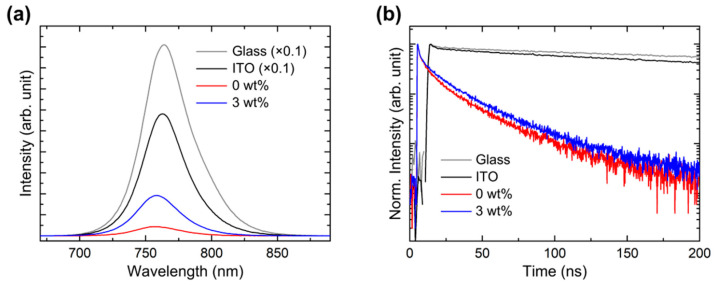
(**a**) Steady-state and (**b**) time-resolved PL spectra of perovskite films on glass, ITO, pristine PEDOT:PSS (0 wt%)/ITO, and 3 wt% Brij C10-mixed PEDOT:PSS/ITO.

**Table 1 polymers-15-00772-t001:** Solar cell parameters of PSCs with Brij C10-mixed PEDOT:PSS HTL.

Concentration	J_SC_ (mA cm^−2^)	V_OC_ (V)	FF (%)	PCE (%)
0 wt%	17.75 (17.57 ± 0.27)	0.85 (0.86 ± 0.01)	53.2 (52.33 ± 0.92)	8.05 (7.90 ± 0.11)
1 wt%	18.62 (18.07 ± 0.34)	0.90 (0.90 ± 0.01)	59.7 (57.50 ± 1.32)	10.00 (9.36 ± 0.23)
2 wt%	19.08 (18.98 ± 0.44)	0.91 (0.91 ± 0.01)	60.2 (58.45 ± 1.20)	10.45 (10.14 ± 0.26)
3 wt%	20.61 (20.67 ± 0.56)	0.89 (0.92 ± 0.03)	62.0 (58.59 ± 2.01)	11.40 (11.07 ± 0.29)
4 wt%	19.51 (19.52 ± 0.59)	0.90 (0.91 ± 0.01)	57.3 (54.83 ± 2.80)	10.04 (9.71 ± 0.17)
5 wt%	19.26 (19.05 ± 0.42)	0.91 (0.88 ± 0.02)	50.0 (50.67 ± 2.23)	8.80 (8.49 ± 0.44)

**Table 2 polymers-15-00772-t002:** Fitting parameters of time-resolved PL spectra of perovskite films on glass, ITO, pristine PEDOT:PSS (0 wt%)/ITO, and 3 wt% Brij C10-mixed PEDOT:PSS/ITO.

	A_1_	τ_1_ (ns)	A_2_	τ_2_ (ns)	A_3_	τ_3_ (ns)	τ_avg_ (ns)
Glass	0.16	0.42	0.57	27.47	99.27	464.92	464.77
ITO	0.40	1.64	9.04	118.93	90.57	378.06	370.16
0 wt%	5.21	1.14	32.49	7.90	62.30	30.18	27.43
3 wt%	5.27	1.28	31.04	10.97	63.69	33.48	30.30

## Data Availability

The data presented in this study are available on request from the corresponding author.

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
