# Peer review of "Enhancement in Power Conversion Efficiency of Perovskite Solar Cells by Reduced Non-Radiative Recombination Using a Brij C10-Mixed PEDOT:PSS Hole Transport Layer"

_polymers, 2023, doi:10.3390/polym15030772_

Round 1
Reviewer 1 Report
Hyunbok Lee et al., “Enhancement in power conversion efficiency of perovskite solar cells by reduced non-radiative recombination using Brij C10-mixed PEDOT:PSS hole transport layer”, demonstrated an increase in PCE PSC, which is due to a decrease in non-radiative recombination by boundary between PEDOT:PSS and perovskite insulating Brij C10. This paper is a rather interesting article that uses appropriate approaches to suppress interfacial recombination, which may affect the future strategy for the development of PSC. However, before any decision is made on its publication, mandatory revision is required in order to clarify some points and increase its attractiveness to the general public. See comments below.
1) How many samples were evaluated?
2) Can you identify the size and architecture of the structures of the same?
3) What is the error associated when evaluating structures processed in the same batch but in different spatial locations?
4) What are the errors associated to structure performances from batch to batch?
5) Was the thickness controlled from batch to batch?
6) Were the conditions used to make the different perovskites the same?
7) The paper shows the results of the UPS and XPS spectra, as well as AFM and SEM images. However, I would recommend adding an additional XRD to the job.
Reviewer 2 Report
The manuscript contains some incremental knowledge related to minor changes in the configuration of the interface between the charge transport and perovskite light absorption layers by adding Brij C10 to the PEDOT:PSS hole transport layer. The scientific novelty of the study for a polymer scientist is quite low, so the study is not likely to attract many readers.
The abstract is too vague and non-quantitative, is should specify for example the range of the study (the concentrations of Brij C10 used) instead of saying that a large amount was used.
The discussion is centered almost entirely on the description of experimental results and fails short of explaining the rationale of choosing Brij C10 over other compounds and the mechanistic behind enhanced performance, compared to other non-conductive polymers. My recommendation to the authors is to add more scientific insight that could be useful for other researchers in the field.
Overall, the study is solid and well presented, except for the SEM images, which should be improved, as the grain boundaries can hardly be seen. The study has no fundamental flaws, is well written and scientifically sound. My main hesitation is that Polymers is probably not the most appropriate journal for this manuscript. Said that, the submission is within the scope of the Special Issue 'Advances in Polyelectrolytes' , so it could be published with minor corrections.
